# Inference of Molecular Regulatory Systems Using Statistical Path-Consistency Algorithm

**DOI:** 10.3390/e24050693

**Published:** 2022-05-13

**Authors:** Yan Yan, Feng Jiang, Xinan Zhang, Tianhai Tian

**Affiliations:** 1School of Mathematics and Physics, Wuhan Institute of Technology, Wuhan 430205, China; yanyan@wit.edu.cn; 2School of Statistics and Mathematics, Zhongnan University of Economics and Law, Wuhan 430073, China; fjiang@zuel.edu.cn; 3School of Mathematics and Statistics, Central China Normal University, Wuhan 430079, China; 4School of Mathematics, Monash University, Melbourne 3800, Australia

**Keywords:** molecular regulation, complex network, graphic model, path consistency, statistical inference

## Abstract

One of the key challenges in systems biology and molecular sciences is how to infer regulatory relationships between genes and proteins using high-throughout omics datasets. Although a wide range of methods have been designed to reverse engineer the regulatory networks, recent studies show that the inferred network may depend on the variable order in the dataset. In this work, we develop a new algorithm, called the statistical path-consistency algorithm (SPCA), to solve the problem of the dependence of variable order. This method generates a number of different variable orders using random samples, and then infers a network by using the path-consistent algorithm based on each variable order. We propose measures to determine the edge weights using the corresponding edge weights in the inferred networks, and choose the edges with the largest weights as the putative regulations between genes or proteins. The developed method is rigorously assessed by the six benchmark networks in DREAM challenges, the mitogen-activated protein (MAP) kinase pathway, and a cancer-specific gene regulatory network. The inferred networks are compared with those obtained by using two up-to-date inference methods. The accuracy of the inferred networks shows that the developed method is effective for discovering molecular regulatory systems.

## 1. Introduction

A molecular regulatory network is a collection of molecular regulators that interact with each other and with other substances in the cell to govern the functions of mRNA molecules and proteins. Experimental studies in recent years have produced a large amount of high-throughout datasets for measuring the gene expression levels or protein activities in the genome scale. Among them, the microarray gene expression data and RNA-sequence data provide rich information to reconstruct genetic regulatory networks, and proteomic data give opportunities to find the protein–protein interactions in cell signaling transduction pathways [1,2]. To study these molecular systems with complex regulations, network science is a powerful tool to investigate the structure of networks and regulatory mechanisms [3,4]. The reverse-engineering study, which is designed to develop genetic regulatory networks or protein–protein interaction networks, is one of the challenging research topics in systems biology and molecular sciences [5,6,7]. In recent years, the advances of single-cell technologies have provided both opportunities and substantial challenges for the development of molecular regulatory networks using single-cell data [8,9,10].

To meet the high demand from biological studies, a wide range of inference methods have been designed to reconstruct regulatory networks based on experimental datasets. There are three major types of inference algorithms according to the mathematical methods used in these algorithms, namely, the correlation-based methods, mechanistic methods, and machine learning methods [11,12,13,14]. The majority of the correlation-based methods use statistical measures or information theory methods to calculate the relationship between each pair of genes or proteins. Due to the efficiency in computation, these algorithms can be used to reconstruct large-scale molecular systems. A number of statistical measures have been employed in these methods, including the Pearson correlation coefficient, Spearman rank correlation, Kendall rank correlation, partial correlation, and distance measures [15,16,17]. The correlation-based methods have also been used to explore the relationship between various types of molecules in both healthy and disease cells [18,19]. A recent study used 213 single-cell datasets to evaluate the performance of 17 association measures [20]. This study suggests that a few association methods do not obtain accurate results for certain datasets in either single-cell or bulk transcriptional datasets. In addition, since time delay is an important issue in gene expression, correlation methods based on time-delay have also been proposed to explain the time differences in gene expression [21,22].

Compared with the correlation coefficients, the mutual information is able to quantify the nonlinear relationship between pairs of variables in the system [23], and thus provides an alternative approach to measure the correlation relationship. A number of algorithms have been designed to reconstruct networks models in biology and financial sciences [24,25,26]. Other measures from the information theory, such as the conditional mutual information and part mutual information [27,28], are able to search for the joint regulations based on the concepts of conditional dependency between a subset of variables. The combination of mutual information and conditional mutual information is able to detect false positive interactions.

The mechanistic methods use mathematical models to simulate the dynamics of molecular behavior. Thus, the developed models can be used to make testable predictions. The Boolean models use binary state vectors to describe state transition trajectories that are governed by a network with the Boolean logic functions [29,30]. To provide the detailed regulatory functions, ordinary differential equation (ODE) models are the widely used approach to describe the continuous changes of molecular dynamics [31,32,33,34]. The linear ODE model is the first option to simulate the dynamics of gene expression [35,36]. In addition, stochastic models, such as the Ornstein–Uhlenbeck (OU) process and Markov processes, have been used to model differential processes [37,38]. Generally, the model-based methods are capable of studying relatively small-scaled networks due to the computational costs and complexity of parameter space.

To reduce the complexity, the hybrid approaches, which combine the correlation-based methods and model-based methods together, are used to infer the gene regulatory networks. The correlation-based methods are employed first to generate sparse networks that are the basis for the next step to use the model-based methods [39,40,41,42,43]. In recent years, there has been a trend to use machine learning techniques for developing genetic regulatory networks [44,45,46].

The path-consistency algorithm (PCA) combines mutual information and high-order mutual information to infer regulatory networks [24,27,28]. Since the conditional mutual information (CMI) needs other variables to determine the correlation relations, PCA is order dependent, namely, the generated network is based on the order of variables in the dataset [47]. To address this issue, the path-consistency (PC) stable algorithm was designed to remove the effect of order dependence [48]. However, this algorithm uses all of the possible CMI in the network and thus, it may increase the false positive rate. In addition, the part mutual information (PMI) is a more accurate measure than the conditional mutual information [28], but it has not been fully considered in the PC-stable algorithm.

When calculating the CMI of genes *X* and *Y*, it requires a third gene *Z* that has a a high correlation relationship to both *X* and *Y*. However, if one of these correlation relationships is removed from the network before the computation, this CMI does not exist. Thus, the path-consistency algorithm depends on the order of variables in the system. If changing the variable order, we may infer a different network using the same algorithm and same experimental dataset. To address the issue of dependence of variable order, this work designs a new algorithm, called the statistical path-consistency algorithm (SPCA), to develop regulatory networks. Rather than using one single variable order to develop the regulatory network, we propose to generate multiple variable orders and then develop a number of different networks. According to the weights of each edge in all these generated networks, we propose the measures to select the final edges. The proposed algorithm is rigorously validated using six golden benchmark networks in DREAM challenges, the mitogen-activated protein (MAP) kinase pathway, and a cancer-specific gene regulatory network.

## 2. Methods

In this section, we first briefly introduce the information theory for measuring the dependent relationship between pairs of genes. The detailed formulas are given in the Appendix A. Then we introduce the path-consistency algorithm. To improve the accuracy, we propose the statistical path-consistent algorithm (SPCA) with part mutual information to estimate the regulatory relationships between genes and proteins.

### 2.1. Information Theory

Mutual information (MI) is designed to measure the dependent relationship between two random variables. Unlike the correlation coefficient that can only measure the linear relationship between random variables, MI is able to describe the nonlinear dependency between random variables. Let the joint density function of two random variables (X,Y) be p(x,y), and the marginal density functions of random variables *X* and *Y* be p(x) and q(y), respectively. MI can be calculated by
(1)$$MI\left(X,Y\right)=-\int \int_{\Omega_{X}\times \Omega_{Y}}p\left(x,y\right)\log\frac{p(x,y)}{p(x)q(y)}dxdy.$$
where $\Omega_{X}$ and $\Omega_{Y}$ are the integral regions of *X* and *Y*, respectively. For application problems, we normally have the observed samples $\left\{\left(x_{1},y_{1}\right),\ldots ,\left(x_{n},y_{n}\right)\right\}$ of the random variables (X,Y). We may use the samples to estimate the density functions, and then use the discrete form of (Equation 1) to calculate the MI value, which is the widely used bin method [25]. We may also assume that random variables (X,Y) follow a particular distribution (e.g., the Gaussian distribution) and then use Formula (Equation 1) to calculate the MI value directly. In this method, the sample data are used to estimate the key parameters in the distribution functions. Alternatively, MI can be obtained by using the values of entropies. The detailed formulas can be found in the Appendix A.

Although a larger value of MI suggests that two random variables may have a closer relationship, for networks with a large number of random variables, the close relationship of two random variables may be based on the strong relationship to the third random variable. CMI is designed to find the conditional relationship between two random variables *X* and *Y*, given the third random variable *Z*, defined by
(2)$$\mathrm{CMI}\left(X,Y|Z\right)=\int \int \int_{\Omega_{X}\times \Omega_{Y}\Omega_{Z}}p\left(x,y,z\right)\log\frac{p(x,y|z)}{p(x|z)q(y|z)}dxdydz,$$
where p(x,y,z) is the joint density function of these three random variables, p(x|z) and q(y|z) are the conditional density functions of random variables *X* and *Y* under the condition of *Z*, respectively, and p(x,y|z) is the conditional density function of (X,Y) given *Z*. In addition, $\Omega_{X}$, $\Omega_{Y}$ and $\Omega_{Z}$ are the integral regions of *X*, *Y* and *Z*, respectively.

When the values of p(x|z) and/or q(y|z) are very small, CMI may be sensitive to small perturbations in the dataset. To address this issue, PMI is proposed to replace CMI for measuring the dependency relationship [28,49]. The key difference between CMI and PMI is that the conditional density functions p(x|z) and q(y|z) in CMI are replaced by partial dependence functions $p^{*}\left(x|z\right)$ and $q^{*}\left(y|z\right)$ in PMI, respectively. For example, function $p^{*}\left(x|z\right)$ in PMI is defined by
$$p^{*}\left(x|z\right)=\sum_{y}p\left(x|z,y\right)p\left(y\right).$$
where p(x|z,y) is the conditional density of *X* given (Y,Z). The detailed formulas for calculating PMI are given in the Appendix A.

### 2.2. Path-Consistency Algorithm

In this work, a gene regulatory network is represented by a graph model G(V,E), where *V* is a set of nodes (i.e., gene), where the size is denoted as |V|=p, and *E* is a set of edges, where each edge e(i,j) connects genes *i* and *j*. The PC algorithm starts from the fully connected network and then removes edges that have relatively weaker independent relationships based on the selected threshold value.

In the first step, MI is used to remove edges from the fully connected network. If the MI value of an edge e(i,j) is less than the threshold value $\epsilon_{1}$, it is assumed that gene *i* and gene *j* are independent, and this edge is removed from the network. After the first stage, we obtain a network whose density should be larger than the desired density. Since PMI is not used in this step, the derived network is called the zero-order PMI network.

In the second step, the first-order PMI is used to remove edges from the zero-order PMI network. For an edge e(i,j) of the zero-order PMI network, if we cannot find any gene that has edges connecting to both gene *i* and gene *j* in the zero-order PMI network, we keep edge e(i,j) in the network. If we can find one or more genes, we calculate the first-order PMI values and keep edge e(i,j) in the network only when the maximal value of these PMI values is larger than the threshold value $\epsilon_{2}$.

We can also apply high-order PMI to further remove edges from the network. For example, when applying the second-order PMI to edge e(i,j), two genes should exist, and each gene has edges that connect to both gene *i* and gene *k*. If these two genes do not exist, edge e(i,j) will remain in the network. Otherwise we calculate the value of the second-order $\mathrm{PMI}(i,j|k_{1},k_{2})$ and keep edge e(i,j) in the network when this PMI value is larger than the threshold value $\epsilon_{3}$. Normally, we use a single threshold value for PMI with different orders. The following Algorithm 1 summarizes the computational step of the PC-stable algorithm using PMI.
**Algorithm 1:** PC algorithm using PMI (PCA-PMI).1:Input: dataset *D* for a network with *N* genes. Set ϵ for deciding the independence.2:Generate a complete network represented by a matrix $G_{N\times N}$ whose elements all are 1.3:Stage L=0 to get a zero-order PMI network. For each edge e(i,j) in network G, calculate the value of MI(i,j) that is zero-order PMI. If MI(i,j)<ϵ, let $G_{ij}=0$ and delete edge e(i,j) from the network. Otherwise keep $G_{ij}=1$.4:Let L=L+1 to get the *L*th-order PMI network.
For each edge e(i,j) in network G derived from the previous stage, find genes that are connected to genes *i* and *j*. Let *T* be the number of all such genes.If T<L, we cannot calculate the *L*th-order PMI and keep this edge in the network.If T>L, selected *L* genes from these *T* genes (i.e. the selection number is $C_{T}^{L}$). For each selection, calculate the *L*th-order PMI.Find the maximal value PMI${}_{\max}$ of all $C_{T}^{L}$ values of the *L*th-order PMI. If PMI${}_{\max}<\epsilon$, let $G_{ij}=0$. Otherwise keep $G_{ij}=1$.If *L* is less than the selected order, go to Step 4 and continue the computation. Otherwise stop the program and output the inferred network *G*.


### 2.3. Statistical Path-Consistency Algorithm (SPCA)

A key issue of the PC algorithm is that the developed network is associated with the order of variables in the system. When PMI${}_{\max}\left(i,j|k\right)<\epsilon$, edge e(i,j) should be deleted in Step 4.4 of Algorithm 1. However, if the edge e(i,k) or e(j,k) is deleted before computing PMI(i,j|k), this PMI does not exist, and edge e(i,j) will remain in the network. To address this issue, the PC-stable algorithm will examine all the possible PMI first. Rather than removing one edge immediately after calculating a particular PMI, the PC algorithm saves the indexes of the edges that should be removed first but still retains the edges in the network. After all the PMI values are determined at one stage, the algorithm removes all the edges that should be removed [48]. In this way, the influence of variable order can be reduced to the minimum.

Since the PC algorithm keeps all the possible edges at one stage, this algorithm may increase the probability of obtaining large PMI values. In addition, the PC algorithm uses the maximal value of all the related PMI values as the final value. Although one large value of PMI is enough to keep that regulation in the network, this method may increase the maximal value and lead to false positive regulations.

To address these issues, we propose a new method to infer regulatory networks. Since the order of variables may be simply determined by the experimental conditions (e.g., the alphabetical order of gene/protein names), we may not be able to obtain accurate inferred networks by using this order. We use random samples to generate a large number of different variable orders. For each generated order, we infer a regulatory network by using Algorithm 1 with a given threshold value. In this way, *N* networks are inferred by using these *N* variable orders. For each edge e(i,j) connecting genes *i* and *j*, there are three possibilities for the appearance of this edge in the inferred *N* networks.

Case 1: Edge e(i,j) appears in all the inferred *N* networks.Case 2: Edge e(i,j) appears in part of the networks but disappears in the other networks.Case 3: Edge e(i,j) disappears in all the inferred *N* networks.

Note that the threshold value in Algorithm 1 should be selected in order that the edge number in Case 1 is less than the expected edge number in the inferred network, but the total edge numbers in Cases 1 and 2 should be larger than the expected edge number. Thus, the key issue is how to rank and select the edges from Case 2.

For any edge e(i,j) in the *k*-th network (k=1,…,N), a weight $w_{ij}^{\left(k\right)}$ is defined based on the appearance of this edge in the *k*-th network, namely $w_{ij}^{\left(k\right)}=1$ if this edge appears in the network, or $w_{ij}^{\left(k\right)}=0$ if this edge is not in the network. We define the mean weight for edge e(i,j) as
(3)$${\mathrm{MW}}_{ij}=\frac{1}{N}\sum_{k=1}^{N}w_{ij}^{k},$$
which is the first criterion to select the edges for the final network. An edge will remain in the network if this weight is larger than the given threshold value.

The mean weight (Equation 3) may be sensitive to the given threshold value ϵ in Algorithm 1. Thus, we considered the second measure that is based on the average of PMI values, defined by
(4)$${\mathrm{APMI}}_{ij}=\frac{1}{N}\sum_{k=1}^{N}{\mathrm{PMI}}_{ij}^{\left(k\right)}w_{ij}^{\left(k\right)},$$
where ${\mathrm{PMI}}_{ij}^{\left(k\right)}$ is the PMI value of edge e(i,j) in the *k*-th network. An alternative approach is to consider the maximal PMI value of corresponding values in all inferred networks, given by
(5)$${\mathrm{MPMI}}_{ij}=\max_{k=1,\ldots ,N}\left\{{\mathrm{PMI}}_{ij}^{\left(k\right)}\right\}.$$

We will test the effectiveness of these three criteria in the following studies. Figure 1 gives the diagram of the proposed SPCA with detailed description. The following Algorithm 2 gives the major steps of SPCA.
**Algorithm 2:** Statistical path-consistent Algorithm (SPCA).**Input**: Dn×m: molecular (gene or protein) activity dataset with *n* variables and *m* observations.V={v1,…,vn}: node set. *N*: number of different variable orders. ϵ: threshold value.
*M*: number of edges in the output network.
**Output**: Network G(V, E). *E* is the set of selected edges.
1:**for** id = 1: N **do**2:   Generate a sample order $\left\{k_{1},\ldots ,k_{n}\right\}$ for {1,…,n} and reorganize the dataset based on the new order of variables $\left\{x_{k_{1}},\ldots ,x_{k_{n}}\right\}$.3:   Use the PCA-PMI algorithm (Algorithm 1) to construct a network based on the new order.4:   For each edge e(i,j), find the PMI value of the highest order ${\mathrm{PMI}}_{ij}^{id}$ (or the edge weight $w_{ij}^{id}$ based on the threshold value ϵ).5:**end for**6:Calculate the value of ${\mathrm{MW}}_{ij}$ (Equation 3), ${\mathrm{MPMI}}_{ij}$ (Equation 4) or ${\mathrm{MPMI}}_{ij}$ (Equation 5). Sort all edges according to the calculated values.7:Select the top *M* edges with the highest values of mean part mutual information or mean weight, and form the regulatory network.8:Export the generated network G(V, VE).


To test the influence of threshold value on the inferred network structure, we use *N* different variable orders to obtain *N* different networks. We consider the maximal, minimal and average edge numbers of these *N* networks based on the various threshold values. The variation ratio is defined by
(6)$$\mathrm{Variation ratio}=\frac{\mathrm{max edge number}-\mathrm{min edge number}}{\mathrm{average edge number}}.$$

### 2.4. Accuracy Measures

The sensitivity and accuracy (ACC) are used in this work to measure the accuracy of inferred networks, which is defined by
$$\mathrm{ACC}=\frac{\mathrm{TP}+\mathrm{TN}}{\mathrm{TP}+\mathrm{FT}+\mathrm{TN}+\mathrm{PN}},$$
and these values are the numbers of the true positive (TP), false positive (FP), true negative (TN) and false negative (FN) in the generated network. To show the accuracy of the methods, we use the receiver operating characteristic curve (ROC) and the area under the ROC curve (AUC) as the measures. In this case, we are interested in the true positive rate (TPR), namely, the proportion of correctly predicted regulations, and false positive rate (FPR), namely the proportion of wrongly predicted regulations, given by
$$\begin{array}{ccc} \mathrm{TRP} & = & \frac{\mathrm{TP}}{\mathrm{TP}+\mathrm{PN}},\\ \mathrm{FPR} & = & \frac{\mathrm{FP}}{\mathrm{TP}+\mathrm{PN}}. \end{array}$$

We also use the positive predictive value (PPV), also called precision, to describe the accuracy of the methods, defined by
(7)$$\mathrm{PPV}=\frac{\mathrm{TP}}{\mathrm{TP}+\mathrm{FP}}.$$

The ideal value of the PPV, with a perfect test, is 1 (100%), and the worst possible value is zero. In addition, we use the harmonic mean of precision and sensitivity (i.e., F1-score) to measure the accuracy of prediction, defined by
(8)$$F1=\frac{2\mathrm{TP}}{2\mathrm{TP}+\mathrm{FP}+\mathrm{FN}}.$$

### 2.5. Experimental Datasets

We use three datasets to test the accuracy of the proposed new algorithm. The first dataset comes from the DREAM3 and DREAM4 challenges. The DREAM3 dataset has the gene expression levels of the SOS DNA repair system [50,51]. This dataset includes 100 genes, and each gene has 100 observations. The exact network includes 166 gene-gene connections. The DREAM4 data consist of in silico networks of gene expression measurements of steady-state levels, obtained by applying 100 different multifactorial perturbations to the original network with, in total, 100 genes [52]. The brief information of the five standard networks from DREAM4 is given in Table 1. For each network, the degree of each gene varies substantially, from 1 to up to 36. In addition, the density of each network is quite low.

The second dataset is for the mitogen-activated protein (MAP) pathway, including the ERK, JNK and p38 pathways, and is one of the most important pathways that regulate a wide range of cellular functions [53,54]. A recent proteomics study generated peptides to almost 12,000 distinct proteins by using 40 breast cancer lines and 4 primary breast tumors [55]. To generate a relatively small dataset for network analysis, we select proteins whose functions are connected to cell proliferation. We use the pathway maps in Kyoto Encyclopedia of Genes and Genomes (KEGG) [56] to select 57 proteins in the MAP kinase pathway. These proteins are the important components of the MAP kinase pathway as well as the crosstalk between the MAP kinase pathway and the Ras and P13K pathways [57]. However, part of these 57 proteins have about a half or even more than a half of missing values. Thus, we use the regularized iterative PCA algorithm in the R package missMDA to estimate the missing values first to generate a complete dataset for these 57 proteins.

The third dataset is the RNA sequencing data for acute myeloid leukemia (AML) based on a large cohort of AML patients from TCGA (http://cancergenome.nih.gov/, access on 23 May 2020) [27,58,59]. The Level-3 processed data are used in this study. The RPKM (read per kilobase of exon per million mapped reads) values are used as the gene expression data. This dataset has the expression levels of 81 genes, including 16 transcriptional factors and 65 target genes. These transcriptional factors include a number of well-known tumor driver genes, such as c-Fos, PU.1 and Egr-1.

## 3. Results

### 3.1. Dependence of Network Structure on Variable Orders

We first examine the dependence of developed networks on the gene order in the regulatory network using the DREAM3 dataset [50,51]. To examine the network structure dependence, we use random samples to obtain 1000 different gene orders and use Algorithm 1 to develop 1000 networks based on the sampled gene orders. Table 2 provides the maximal, minimal and average edge numbers of these 1000 networks based on different threshold values. It shows that the developed networks are dependent on the generated gene orders. For the four tests in Table 2, the variation ratios all are above 10%. Among them, the largest ratio is 17.14%. In addition, the ratio value is related to the total number of edges in the network. If the edge number is smaller, the ratio is larger.

Figure 2 gives the frequency for the edges of Case 2, namely edges appearing in only part of the networks, in all 1000 networks, using four different threshold values. In all four tests, the edge numbers are evenly distributed. There are about 50% of Case 2 edges that appear in more than a half of the generated 1000 networks. In addition, the edge number with frequency ∼0.5 is relatively large, which increases the difficulty for selecting edges. The edge number of Case 2 increases in accordance with the increase in total network edge number by using a small threshold value. For example, the edge number of Case 2 is 104 when ϵ=0.05. However, it increases to 155 when ϵ=0.016.

### 3.2. Effectiveness of SPCA

After examining the dependence of network structure on the variable order, we next show the accuracy of the proposed SPCA by comparing with the two published methods, namely the PCA-PMI and PC-stable algorithm with PMI. We first use the SOS DNA repair gene dataset for reconstructing gene regulatory networks [51]. We use the published PCA-PMI algorithm in the literature [24] and the R-package pcalg for the PC-stable method [60]. Figure 3 shows that SPCA has an AUC value of 0.8649 that is larger than the value of the PC-stable algorithm with PMI (i.e., 0.8605) and that of PCA-PMI (i.e., 0.8571). These results suggest that SPCA has better accuracy than these two algorithms. In addition, the AUC value of the PC-stable algorithm is slightly smaller than that of the PCA-PMI, which suggests that the PC-stable algorithm may increase the false–positive regulations.

We next use five datasets from the DREAM4 challenge to further investigate the accuracy of our proposed algorithm. We apply the three methods, namely the PCA-PMI, PC-stable algorithm with PMI, and our SPCA algorithm, to infer the five gene networks. For the SPAC algorithm, we test three criteria for selecting the putative regulations. For each network, we run the algorithms with five threshold values (ϵ=0.031∼0.035) and obtain the networks with slightly more edges than that of the true network. The ranges of edge numbers of all the inferred networks are given in Table 3. Figure 4 gives the AUC values of these three methods applied to networks 1∼4. The averaged AUC values in Table 3 suggest that our proposed SPCA algorithm achieves better accuracy than the other two methods. In addition, the maximal PMI criterion achieves better accuracy than the other two average criteria. Note that the accuracy of the PC-stable algorithm is not as good as that of the PCI-PMI algorithm for some networks. In addition, the performance of SPCA with the average PMI criterion (Equation 3) is not as good as the other criteria, but the SPCA with the maximal PMI criterion (Equation 5) has the best performance among the three proposed criteria.

Table 4 and Table 5 give the FFV values and F1-scores of these three methods applied to networks 1∼5. Both the FFV values and F1-scores suggest that our proposed SPCA algorithm with the maximal PMI criterion (Equation 5) achieves better accuracy than the other two methods. In addition, the maximal PMI criterion achieves better accuracy than the other two average criteria. Note that the accuracy of the PC-stable algorithm is not as good as that of the PCI-PMI algorithm for some networks. In addition, the performance of SPCA with the mean weight (Equation 3) and average PMI criteria (Equation 4) is not as good as the other two methods, but the SPCA with the maximal PMI criterion (Equation 5) has the best performance among the three proposed criteria. Figure 5 gives the FFV values and F1-scores of these three methods applied to networks 1∼3. When networks have more edges, the FFV values and F1-scores are smaller. The results in Figure 5 are consistent with those shown in Figure 4. However, if the AUC values of the SPCA algorithm with mean weight (Equation 3) or average PMI (Equation 4) are just marginally better than those of the other two methods, the corresponding PPV value and/or F1-score of the SPCA algorithm may not be as good as those of the other two methods for certain datasets.

The computing time is not an issue for the implementation of the proposed method. The CPU time of PCA-PMI is only 5 s for a network with 100 genes using an Apple iMac with 3.4 GHz processor. Although the required time of the proposed method is *N* times the computing time of PCA-PMI, the value of mutual information is the same for different variable orders, which can be used to reduce the computing time. To infer a network with 100 genes using 1000 different variable orders, the computing time is less than 5 min using the same computer.

### 3.3. Map Kinase Network

The MAP kinase cascade includes a number of kinases that transfer cellular signals from the trans-membrance protein receptor on the cell surface to DNA in the nucleus to regulate cellular functions [53,54]. We next apply the proposed SPCA to infer the regulatory structure of the MAP kinase pathway. By using different threshold values, we develop three networks with ∼57, ∼114 and ∼170 edges. When the number of edges is ∼57 or ∼114, quite a few proteins form a small subnetwork and are isolated from the main network. To build a connected network, we first use the minimum spanning tree (MST) to connect all proteins first, and then add the edges with the largest PMI weights to the network.

Figure 6 gives the inferred network with 106 connections. We use different colors to represent proteins with different connection numbers (i.e., degree). It shows that proteins in the MAP kinase three-tier modules have relatively larger degrees, such as the proteins Raf1, RafB, and MKK3, that are three MAPK proteins. On the other hand, the downstream target genes may have fewer connections to other proteins, such as transcriptional factors p53 and JunD. However, this rule is not always true. For example, the connection to kinase p38 is only one.

There are four types of connections in the developed network. The first one represents protein regulations that have been confirmed by experimental studies, such as the connections MEK-ERK, PI3K-AKT, and RafB-AKT. The second type is for connections of protein isoforms or proteins that are regulated by the same upstream protein. For example, MEK1-MEK2 connects two isoforms of MEK protein. Similar connections include MKK3-MKK6. Quite a number of connections belong to the third type, namely indirect connections that may have other proteins between these connected proteins. For example, Rafb-ERK is an indirect connection of the MAP kinase module Raf-MEK-ERK. Similar connections include GRB2-Raf1, and MEKK3-MAPKAPK. For the last type of connections, we cannot find any published results to support these connections. These connections are the predictions of putative regulations or may be special connections in cancer cells. Note that the above conclusions are based on the pathway maps from KEGG. In recent years, experimental studies have found more connections between the proteins in the MAP kinase pathway.

### 3.4. Reconstruction of Cancer-Specific Gene Regulatory Network

After the study of the MAP kinase pathway, we next examine a gene regulatory network for acute myeloid leukemia (AML) [27]. Due to the dysfunction of certain functional modules and pathways, the regulatory relationships between transcriptional factors and target genes in the disease cells may be different from those in the healthy cells [61]. To explore the gene regulations in cancer cells, we apply SPCA to develop a regulatory network for AML using the RNA-seq data of a large cohort of AML patients from TCGA [27,58,59]. Figure 7 shows the inferred AML-specific gene network by using SPCA. In this network, there are 81 total cancer genes, including 16 transcriptional factors and 65 target genes. This system has also been studied by RACER [62] to develop a regulatory network using the same AML gene expression dataset. By using different threshold values, we can detect different numbers of regulations. Although there are 227 reported regulations, the real regulations in cancer cells are not known. Here, our inferred network is only a prediction for the potential regulations. Further experimental studies are needed to evaluate these predictions.

## 4. Discussions and Conclusions

The motivation of this study is to address the issue of variable order dependence when using the conditional correlation methods to infer complex systems. An observation dataset has a variable order that is determined by the experimental conditions, for example, the alphabetic order of gene names or the rank of patient ID numbers. The variations of the given variable order lead to different inferred networks that may have better accuracy. Since the optimal order may not exist or is not known, we generate a number of different variable orders, and then calculate the PMI values of each edge based on different variable orders and infer different regulatory networks. The key question is how to derive the final inferred network based on these networks. In this work, we propose three criteria to determine the weight or PMI value of each edge and then infer the final network. The research results suggest that the criterion using the maximal PMI value leads to the inferred networks with better accuracy.

With more and more generated omics datasets, there is a strong need to design effective algorithms to infer regulatory relationships or functional relationships between genes and proteins. However, it is well recognized that the gene expression process is nonlinear, sparse and systematic. In addition, time delay widely exists in gene expression and the length of delay varies from gene to gene. In the current information-theory-based methods, a single threshold value is normally used to select the significant regulations. We tested the DREAM challenge datasets but found certain regulations have quite small correlation coefficient values and MI values. Thus, these regulations may never be selected unless a small threshold value is used. However, in that case, the inferred network would be quite dense. On the other hand, it is relatively easy to incorporate nonlinearity and time delay into a dynamic model, but the inference of model parameters is still a challenging issue for a relatively large network. Although quite a large number of algorithms have been proposed to infer regulatory networks [13,63], the accuracy of developed networks is still not satisfactory. The further research in this area is still challenging and exciting.

In summary, this work develops a new method called SPCA to infer molecular regulation networks using various omics datasets. To address the issue of the dependence of network structure on the variable order, this method generates a number of variable orders and develops a network by using the PCA-PMI algorithm, using each variable order. Information theory is used to measure the correlation relationship between each pair of genes or proteins. We calculate the weight or PMI value of each edge based on the networks using a different variable order, and use a threshold value to select the final putative regulations. The standard networks in DREAM challenges are used to examine the accuracy of the proposed method. We also use the new method to develop the mitogen-activated protein (MAP) kinase pathway, and a cancer-specific gene regulatory network. Our studies suggest that the developed method is effective for discovering molecular regulatory systems. In this work, we propose a general approach to improve the inference accuracy by generating a large number of different variable orders. This approach may be applied to certain current state-of-the-art methods to obtain more promising results. This will be an interesting research topic for our future research.

## Figures and Tables

**Figure 1 entropy-24-00693-f001:**
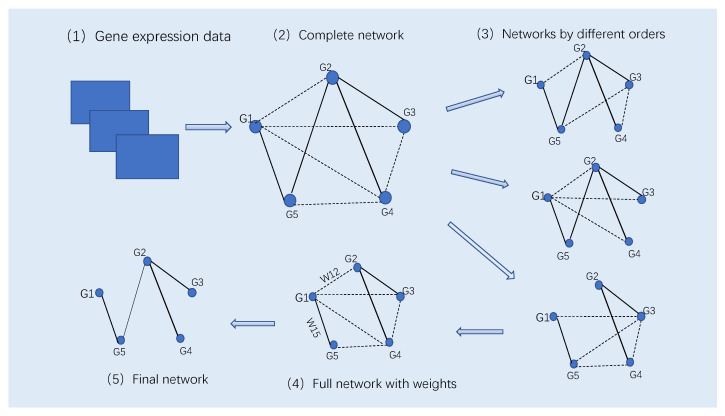
The diagram of SPCA. (1) The omics dataset with *n* genes and *m* observations. (2) The complete network in which each pair of genes are connected. The solid and dash lines in the network represent direct and indirect regulations, respectively. (3) A number of networks are inferred by using the PCA-PMI algorithm (Algorithm 1) based on the generated different variable orders. (4) Calculate the weight of each edge in the complete network based on the corresponding PMI values in all inferred networks in Step (3). (5) Select edges with the largest values of PMI measures to form the inferred network.

**Figure 2 entropy-24-00693-f002:**
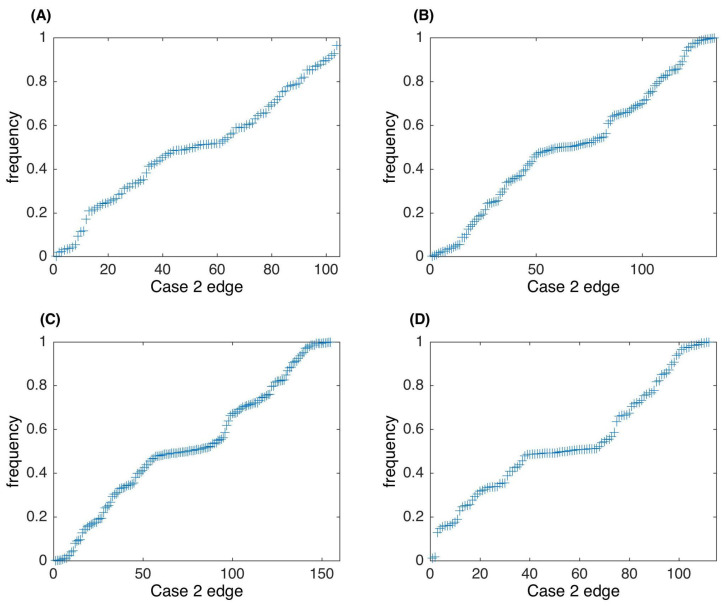
Edge frequency of the Case 2 edges in 1000 networks. (**A**) Threshold value ϵ=0.05. (**B**) ϵ=0.03. (**C**) ϵ=0.02. (**D**) ϵ=0.016.

**Figure 3 entropy-24-00693-f003:**
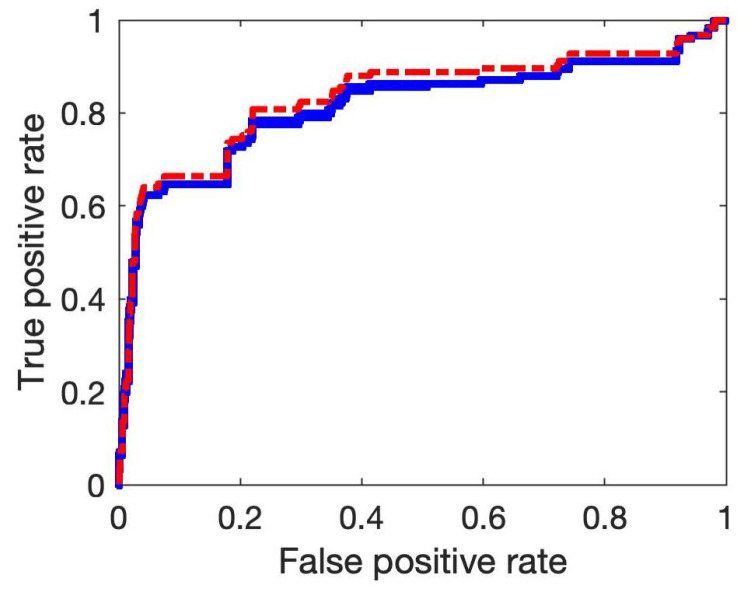
ROC curves of the inferred gene networks by using PCA-PMI (green line), PC-stable-PMI (blue line) and our proposed SPCA (red line). A larger value of AUC shows the method is more accurate.

**Figure 4 entropy-24-00693-f004:**
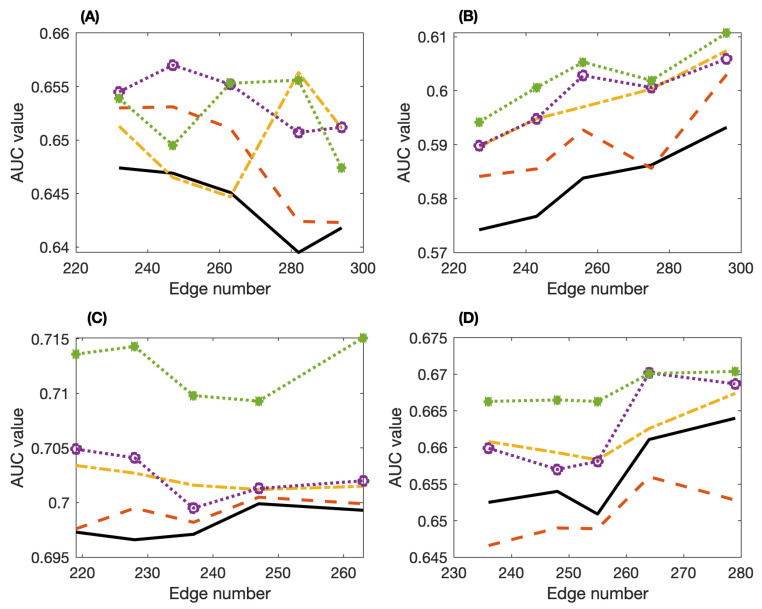
Values of the area under ROC curve (AUC) of the inferred networks for datasets 1∼4 from the DREAM4 challenge by using the PCA-PMI algorithm, PC-stable algorithm with PMI and our proposed SPCA algorithm. (**A**) Dataset 1; (**B**) dataset 2; (**C**) dataset 3; (**D**) dataset 4. (PCA-PMI: solid-line, PC-stable-PMI: dash line, SPCA with mean weight (Equation 3): dash-dot line, SPCA with average PMI (Equation 4): circle, SPCA with maximal PMI (Equation 5): star).

**Figure 5 entropy-24-00693-f005:**
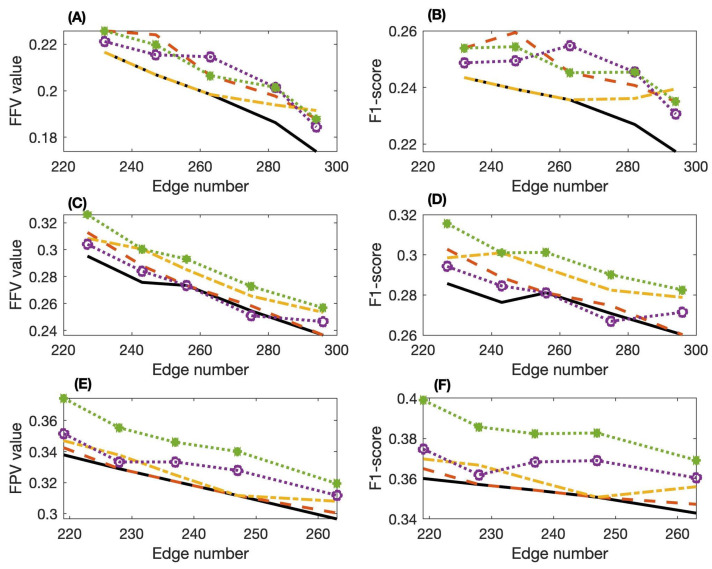
Positive predictive values (PPV) and harmonic mean of precision and sensitivity (F1-score) of the inferred networks for datasets 1∼3 from the DREAM4 challenge by using the PCA-PMI algorithm, PC-stable algorithm with PMI and our proposed SPCA algorithm. (**A**) PPV values of dataset 1; (**B**) F1-scores of dataset 1; (**C**) PPV values of dataset 2; (**D**) F1-scores of dataset 2; (**E**) PPV values of dataset 3; (**F**) F1-scores of dataset 3. (PCA-PMI: solid-line, PC-stable-PMI: dash line, SPCA with mean weight (Equation 3): dash-dot line, SPCA with average PMI (Equation 4): circle, SPCA with maximal PMI (Equation 5): star).

**Figure 6 entropy-24-00693-f006:**
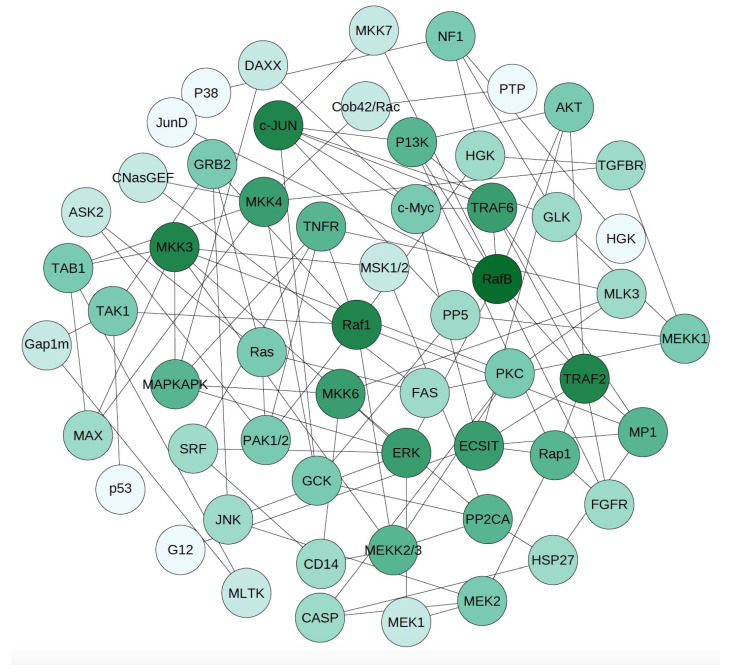
Inferred network of the MAP kinase pathway with N=57 proteins and 106 connections by using the proposed SPCA. A protein with darker color has more inferred connections.

**Figure 7 entropy-24-00693-f007:**
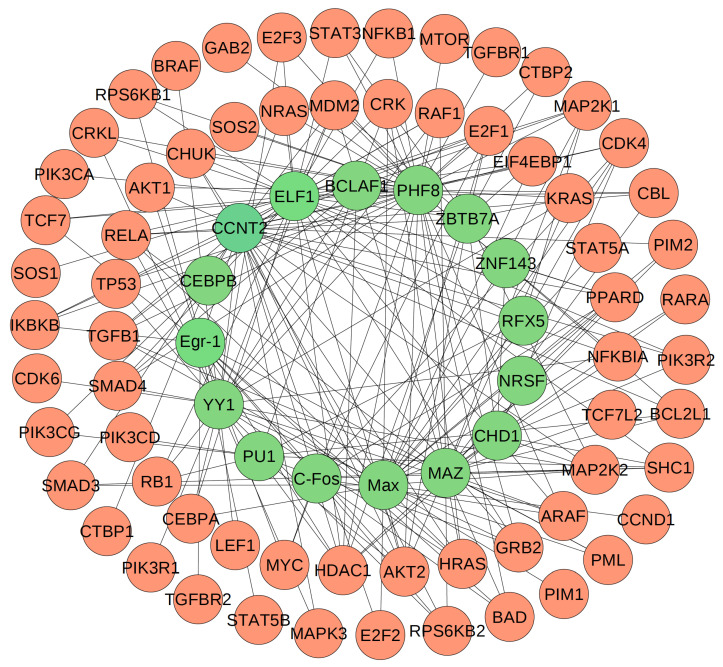
The inferred acute myeloid leukemia (AML) genetic regulatory network generated by SPCA. Blue genes: regulators. Pink genes: target genes.

**Table 1 entropy-24-00693-t001:** Descriptions of the five datasets for gene expression activities from the Dream4 challenge.

Dataset	No. of Genes	No. of Samples	Max Degree	Min Degree	No. of Edges	Density
Dataset 1	100	100	26	1	169	0.03483
Dataset 2	100	100	38	1	242	0.04988
Dataset 3	100	100	16	1	192	0.03957
Dataset 4	100	100	16	1	207	0.04267
Dataset 5	100	100	18	1	191	0.03937

**Table 2 entropy-24-00693-t002:** Variations ratio of the 1000 generated networks using four different threshold values ϵ for the DREAM3 network with 100 genes.

ϵ	Mean Edge Number	Min Edge Number	Max Edge Number	Variation Ratio
0.05	105	95	113	17.14%
0.03	155	136	161	16.13%
0.02	197	186	209	11.68%
0.016	250	234	262	11.12%

**Table 3 entropy-24-00693-t003:** Ranges of the edge numbers and values of the area under ROC curve (AUC) for the inferred networks of the five gene expression datasets from the DREAM4 challenge. (MW: mean weight (Equation 3), APMI: average PMI (Equation 4), MPMI: maximal PMI (Equation 5)).

Dataset	Edge Ranges	PCI-PMI	PC-Stable	SPCA(MW)	SPCA(APMI)	SPCA(MPMI)
1	232∼294	0.6441	0.6484	0.6475	0.6537	0.6523
2	227∼296	0.5828	0.5902	0.5978	0.5988	0.6025
3	219∼263	0.6980	0.6991	0.7021	0.7024	0.7124
4	236∼279	0.6565	0.6507	0.6611	0.6628	0.6679
5	243∼297	0.6902	0.6882	0.6914	0.6919	0.6999

**Table 4 entropy-24-00693-t004:** Ranges of the edge numbers and Positive predictive values (FFV) for the inferred networks of the five gene expression datasets from the DREAM4 Challenge. (MW: mean weight (Equation 3), APMI: average PMI (Equation 4), MPMI: maximal PMI (Equation 5)).

Dataset	Edge Ranges	PCI-PMI	PC-Stable	SPCA(MW)	SPCA(APMI)	SPCA(MPMI)
1	232∼294	0.1964	0.2084	0.2015	0.2074	0.2083
2	227∼296	0.2671	0.2738	0.2826	0.2718	0.2898
3	219∼263	0.3192	0.3208	0.3259	0.3316	0.3470
4	236∼279	0.3064	0.2919	0.3104	0.3041	0.3200
5	243∼297	0.3462	0.3296	0.3505	0.3436	0.3614

**Table 5 entropy-24-00693-t005:** Ranges of the edge numbers and values of the harmonic mean of precision and sensitivity (F1-score) for the inferred networks of the five gene expression datasets from the DREAM4 challenge. (MW: mean weight (Equation 3), APMI: average PMI (Equation 4), MPMI: maximal PMI (Equation 5)).

Dataset	Edge Ranges	PCI-PMI	PC-Stable	SPCA(MW)	SPCA(APMI)	SPCA(MPMI)
1	232∼294	0.2325	0.2468	0.2388	0.2458	0.2468
2	227∼296	0.2671	0.2738	0.2826	0.2718	0.2898
3	219∼263	0.3530	0.3549	0.3605	0.3669	0.3838
4	236∼279	0.3462	0.3296	0.3505	0.3436	0.3614
5	243∼297	0.2964	0.2881	0.2962	0.2946	0.3036

## Data Availability

Data sharing is not applicable to this article.

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
