# Peer review of "Inference of Molecular Regulatory Systems Using Statistical Path-Consistency Algorithm"

_entropy, 2022, doi:10.3390/e24050693_

Round 1

Reviewer 1 Report

This work presents statistical path consistency algorithm to find variable dependence order. The method randomly generates and tests variable orders and infer a network. The method is assessed on benchmark dream datasets.

1. Section 1.1 in methods contains mostly text book level information not appropriate for a research article. I suggest this should be removed or substantially reduced in size.

2. Instead of the text book detail of entropy, and definitions of TPR, FPR, accuracy etc, the authors should dedicate a sections on describing the data

2. The test metric used here is mainly auroc. The measures used here only give part of the picture. The authros should also report F1 scores and PPV to enable the reader to get a more complete picture of the performance of the algorithms. 

3. I do not understand the reason why the authors chose to compare with PCA-PMI and PC-Stable algorithms ans both are fairly dated (these were published 10 years ago in 2012). Surely there are current state of the art methods to accomplish this task. The authors should compare with current state of the art methods.

4. The language of the paper, while accessible, needs some revisions.

Page 1, Line 8. "edge weigth" is repetitive.
Page 2, Line 37. Reference shows up as  "?"
Page 4, typos in the unnumbered line below 116.
Page 5, Line 128. Tt should be "remove"
Page 5, Line 140. It should be "second-order PMI"
Page 5, Line 140. This needs revision "...we need to find two genes  that all are connected to genes i and j ..."
Page 5, Line 144. This sentence is unclear and doesnt make sense in its current form "Note that, when the order of PMI is larger, it is less likely to find the high-order PMI"
Page 10, Table 2: The meaning of the phrases "Gene No", "Sample No', "edge no" is different from  what the authors probably intend to convey here. Instead, it should be "No. of Genes", "No. of Samples"|, and "No. of Edges". 

Author Response

This work presents statistical path consistency algorithm to find variable dependence order. The method randomly generates and tests variable orders and infer a network. The method is assessed on benchmark dream datasets.

Comment 1. Section 1.1 in methods contains mostly text book level information not appropriate for a research article. I suggest this should be removed or substantially reduced in size.

Response: we have rewritten this subsection and reduced the length substantially. Since the other referee is positive about this part, we move the detailed description to Supplementary Information.

Comment 2. Instead of the text book detail of entropy, and definitions of TPR, FPR, accuracy etc, the authors should dedicate a sections on describing the data

Response: we added one subsection "Experimental datasets" and put all the descriptions for three datasets together.

Comment 3. The test metric used here is mainly auroc. The measures used here only give part of the picture. The authros should also report F1 scores and PPV to enable the reader to get a more complete picture of the performance of the algorithms.

Response: We added PPV values and F1 scores to the results section as Tables 4 and 5. Figure 5 is added to provide some detailed values.  

Comment 4. I do not understand the reason why the authors chose to compare with PCA-PMI and PC-Stable algorithms ans both are fairly dated (these were published 10 years ago in 2012). Surely there are current state of the art methods to accomplish this task. The authors should compare with current state of the art methods.

Response: We thank the referee for this very good suggestion. We would like to clarify that the method proposed in this work is a general approach, namely to generate different variable orders in order to find better results. This approach may be applied to some current state of the art methods to get better results as well. Thus, at the end of the Conclusion section, we added the following sentence:

In this work, we propose a general approach to improve the inference accuracy by generating a large number of different variable orders. This approach may be applied to certain current state of the art methods to obtain more promising results. This will be an interesting research topic for our future research. 

Comment 5. The language of the paper, while accessible, needs some revisions.

Response: We have read the manuscript very carefully. We have made changes according to the comments below, and also made a number of other changes to errors and typos.

Page 1, Line 8. "edge weigth" is repetitive.

Response: We thank the referee for all these detailed comments. We removed "using the corresponding edge weights in the inferred networks,"

Page 2, Line 37. Reference shows up as  "?"

Response: we corrected this citation.

Page 4, typos in the unnumbered line below 116.

Response: we changed the writing for this part. It does not appear in the paper now.

Page 5, Line 128. Tt should be "remove"

Response: we made the change.

Page 5, Line 140. It should be "second-order PMI"

Response: we have made the change.

Page 5, Line 140. This needs revision "...we need to find two genes  that all are connected to genes i and j ..."

Response: we changed this sentence to "when applying the second-order PMI to edge e(i, k), two genes should exist, and each gene has edges that connect to both gene i and gene k. "

Page 5, Line 144. This sentence is unclear and doesnt make sense in its current form "Note that, when the order of PMI is larger, it is less likely to find the high-order PMI"

Response: we have removed this sentence.

Page 10, Table 2: The meaning of the phrases "Gene No", "Sample No', "edge no" is different from  what the authors probably intend to convey here. Instead, it should be "No. of Genes", "No. of Samples"|, and "No. of Edges".

Response: We have made changes according to the comment. This table is Table 1 in the revised version.

Reviewer 2 Report

General Comments:

  • Introduction is well written and informative
  • Methods
    • Contains methods and results which I would separate, mixing the results with the discussion would be better.
    • The description of information theory and the two algorithms was helpful.
    • Should create a subsection that talks specifically about the datasets/networks used for testing the algorithms and include a subsection of the accuracy of the algorithms as well. This will make the text easier to follow.
  • Discussion
    • Most of the relevant discussion is found in the methods section, and would be better suited in the discussion section. The authors are highlighting only a few additional points
  • Overall writing
    • See below for some specifics, but a more thorough review should be done for grammar and correct spelling. Words are spelled correct, but the wrong word is used.

Individual Items:

Line 37: Unsure why there is a ? in the references

Line 88: CMI is not spelled out before it’s use

Line 106: “To improving” should be “To improve”

Below line 116: “calculating MI the” should be “calculating MI is the”

Line 145: “summaries” should be “summarizes”

Line 183: “still remains” should be “still retains”

Line 233: “testes” should be “tested”

Author Response

Introduction is well written and informative.
The description of information theory and the two algorithms was helpful.
Discussion
Most of the relevant discussion is found in the methods section, and would be better suited in the discussion section. The authors are highlighting only a few additional points

Methods
Comment 1: Contains methods and results which I would separate, mixing the results with the discussion would be better.

Response: We moved the discussion of computing time to the Results section.

Comment 2: Should create a subsection that talks specifically about the datasets/networks used for testing the algorithms and include a subsection of the accuracy of the algorithms as well. This will make the text easier to follow.

Response: We added one subsection "Experimental datasets" in the Methods section and moved all descriptions of datasets together.

Overall writing
Comment 3: See below for some specifics, but a more thorough review should be done for grammar and correct spelling. Words are spelled correct, but the wrong word is used.

 Response: We thank the referee for these detailed comments. We have made the corresponding changes. We also read the manuscript very carefully and made a number of changes to errors and typos.

Individual Items:
Line 37: Unsure why there is a ? in the references

Response: we have corrected the citation.

Line 88: CMI is not spelled out before it’s use

Response: We added the notation in line 77.

Line 106: “To improving” should be “To improve”

Response: we have made the change.

Below line 116: “calculating MI the” should be “calculating MI is the”

Response: This part was moved to the supplementary information according to the comment of the other referee. This sentence was rewritten.
Line 145: “summaries” should be “summarizes”

Response: We have made the change.

Line 183: “still remains” should be “still retains”

Response: We thank the referee for this comment. We changed all "remain" to "retain".

Line 233: “testes” should be “tested”

Response: we have changed “testes” to “test”

Round 2

Reviewer 1 Report

I am happy with the revised version of the manuscript. 

Author Response

We thank the referee for the encouragement. We have read the manuscript very carefully and made changes regarding typos and writing errors.

Reviewer 2 Report

Thanks for the revision, it was much easier to read and the material will be useful for future work.  I have just a few minor changes related to wording:

In lines 154-155: The wording makes it sound like you retain the edge if the genes don't exist, which from the prior lines when they do exist, you also keep them.  Just want to make sure that don't remove them if the genes don't exist.

Equation 8: Insead of P1, it should be F1

Line 281: ACU should be AUC

Line 299-300: You already spell out AUC so you don't need to again

Line 378: Instead of "In recently year" say "In recent years"

Line 392: Instead of "are totally 81 cancer genes" use "are 81 total cancer genes"

Line 402: Instead of "variables" use "variable"

Finally,  you changed "remains" to "retains" in a number of places, I would suggest if you are talking about the edges directly use remain, "the edge remains" if you are talking about the algorithm use retain, "PC-algorithm retains the edge"

Author Response

Comment 1: In lines 154-155: The wording makes it sound like you retain the edge if the genes don't exist, which from the prior lines when they do exist, you also keep them.  Just want to make sure that don't remove them if the genes don't exist.

Response: We thank the referee to pointing this out. We changed these sentences as

For an edge e(i, j) of the zero-order PMI network, if we cannot find any gene that has edges connecting to both gene i and gene j in the zero-order PMI network, we keep edge e(i, j) in the network. If we can find one or more genes, we calculate the first-order PMI values and keep edge e(i, j) in the network only when the maximal value of these PMI values is larger than the threshold value ε2.

Comment 2: Equation 8: Instead of P1, it should be F1

Response: We made the change.

Comment 3: Line 281: ACU should be AUC

Response: We made the change.

Comment 4: Line 299-300: You already spell out AUC so you don't need to again

Response: We removed the second definition.

Comment 5: Line 378: Instead of "In recently year" say "In recent years"

Response: We made the change.

Comment 6: Line 392: Instead of "are totally 81 cancer genes" use "are 81 total cancer genes"

Response: We made the change.

Comment 7: Line 402: Instead of "variables" use "variable"

Response: We made the change.

Comment 8: Finally, you changed "remains" to "retains" in a number of places, I would suggest if you are talking about the edges directly use remain, "the edge remains" if you are talking about the algorithm use retain, "PC-algorithm retains the edge"

Response: We thank the referee for this very good advice. We made changes according to the suggestion.